# Stress sensitization among severely neglected children and protection by social enrichment

Mark Wade[1]*, Charles H. Zeanah[2], Nathan A. Fox[3], Florin Tibu [4], Laura E. Ciolan [5] & Charles A. Nelson[6,7]

Childhood adversity may sensitize certain individuals to later stress which triggers or amplifies psychopathology. The current study uses data from a longitudinal randomized controlled trial to examine whether severe early neglect among children reared in institutions increases vulnerability to the effects of later stressful life events on externalizing problems in adolescence, and whether social enrichment in the form of high-quality foster care buffers this risk. Children abandoned to Romanian institutions were randomly assigned to a foster care intervention or care-as-usual during early childhood. A sample of never-institutionalized children served as a comparison group. Here we report that, among those with prolonged institutional rearing, more stressful life events in preadolescence predicted higher externalizing problems in adolescence. This effect was not observed for never-institutionalized children or those in foster care, thus providing experimental evidence that positive caregiving experiences protect against the stress-sensitizing effects of childhood neglect on externalizing problems in adolescence.

---

[1] Department of Applied Psychology and Human Development, University of Toronto, Toronto, ON, Canada. [2] Department of Psychiatry and Behavioral Sciences, Tulane University School of Medicine, New Orleans, LA, USA. [3] Department of Human Development and Quantitative Methodology, University of Maryland, College Park, MA, USA. [4] Department of Health and Human Development, Stefan cel Mare University of Suceava, Suceava, Romania. [5] Faculty of Psychology and Educational Sciences, University of Bucharest, Bucharest, Romania. [6] Boston Children's Hospital, Boston, MA, USA. [7] Harvard Graduate School of Education, Cambridge, MA, USA. *email: m.wade@utoronto.ca

Different types of childhood adversity demonstrate little disorder specificity[1], suggesting that experiences such as abuse and neglect are transdiagnostic vulnerability factors for several psychiatric conditions. Among children experiencing profound neglect in the context of institutional rearing, higher rates of both internalizing and externalizing disorders are observed across development[2,3]. The causal impact of neglect on psychopathology is highlighted by the finding that institutionally-reared children who are randomly assigned to foster care early in life have lower internalizing and externalizing problems[2,4], and lower general psychopathology risk[5], compared with children experiencing prolonged institutional rearing.

There is also considerable evidence that less insidious forms of stress associated with relatively common life events are associated with more internalizing and externalizing problems[6,7]. The effects of stressful life events may be especially prominent during the transition to adolescence, a potentially sensitive period in which the brain is acutely vulnerable to stress[8]. In animal models, pubescence appears to be a transitional window through which stress has long-term consequences for psychopathology[9]. In humans, it is well recognized that puberty is accompanied by substantial social and biological change, with implications for later psychopathology[10]. For example, more than 70% of mental health problems in adulthood have their onset in childhood or adolescence[11], and stress during adolescence is believed to play a critical role in the pathophysiology of mental health problems later in life[12].

Despite the significant association between stressful life events and psychopathology in adolescence, not all individuals who experience such stressors will develop psychiatric disorders. One hypothesis to explain this inter-individual variability is that risk for psychopathology is heightened among those who have also experienced childhood adversity. In other words, childhood adversity or maltreatment may sensitize individuals to the effects of later stressful life events on psychopathology[13]. This stress sensitization effect has been documented across several studies in both adolescents and adults[14–16]. Under this hypothesis, exposure to childhood adversity lowers the threshold for tolerating future stressful events that trigger the onset of psychopathology or aggravate underlying vulnerabilities. Consistent with the notion that adolescence may be a sensitive period in development, this sensitization effect appears to be stronger for stressful events that occur during adolescence compared with adulthood[17].

The preponderance of previous research examining stress sensitization as a function of childhood adversity has focused on internalizing disorders such as depression and anxiety. There is also some evidence for stress sensitization on intimate partner violence[18], binge drinking[19], and substance use[20], outcomes traditionally included on the externalizing spectrum. However, no study has examined the stress sensitization hypothesis with respect to broadband externalizing problems in adolescence. This is especially important in the context of institutional rearing, as externalizing problems appear more responsive to foster care intervention following institutional care than internalizing problems[2]. Also, after controlling for overlap with other domains of psychopathology, externalizing problems decline over the transition to adolescence among children in foster care, while those with prolonged institutional deprivation show persistent difficulties over this period[5]. This same pattern is not observed for internalizing problems. Moreover, blunting of stress systems is a mediator of the effects of early adversity on externalizing problems[21,22], and chronic stress exposure is more strongly linked to externalizing-specific than internalizing-specific problems after controlling for shared variance[23]. Thus, the first goal of the present study was to examine whether severe early neglect as a result of institutional rearing sensitizes children to the effect of later stressful life events on externalizing problems in adolescence after accounting for covariance with other domains of psychopathology.

The second goal was to examine whether social enrichment protects children against the sensitizing effects of early neglect. In the context of the current study, social enrichment refers to the provision of development-enhancing caregiving behaviors via a foster care intervention that was specifically designed to meet the needs of institutionally-reared children, including enhanced caregiver availability, sensitivity, stimulation, and positive affectivity[24]. The intervention has been shown to enhance attachment security in early childhood[25]. Since the intervention took the form of a randomized controlled trial (RCT; described below), this study provides a rare experimental test of how social enrichment afforded by foster care may safeguard against the stress-sensitizing effects of early neglect on psychopathology in adolescence.

Finally, it may be important to differentiate the effects of dependent and independent life events. Dependent events are those that are at least partially controllable or influenced by the individual (e.g., "got in trouble with the law"), while independent events are those that are outside the individual's control (e.g., "a family member or close relative died"). In adolescence, there is evidence that independent events may be more germane to the precipitation of psychopathology than dependent events among those with a history of adversity[15,17]. Similar results favoring the deleterious effects of independent events have been demonstrated for alcohol consumption in adulthood[20]. On this basis, we hypothesized that early psychosocial neglect would more strongly sensitize children to the effects of later independent events on externalizing problems compared with dependent events.

We show that more stressful life events in preadolescence are associated with higher externalizing problems in adolescence, but only among children with a history of prolonged early deprivation. Among those assigned to early foster care and those without a history of institutional rearing, there was no relation between stressful life events and externalizing problems. These effects were more pronounced for independent as opposed to dependent events.

## Results

**Descriptive results.** Table 1 presents descriptive statistics and bivariate correlations between primary variables, covariates, and secondary variables of interest. As can be seen in this table, the total number of stressful life events at age 12 was positively associated with externalizing problems at both ages 12 and 16. Independent events were more strongly correlated with externalizing problems than dependent events. Males had higher externalizing problems than females at both ages. Neither age when children entered the institutions, nor age when children entered foster care, was associated with any of the primary variables of interest.

Supplementary Fig. 1 presents group differences in total, dependent, and independent life events. In short, the care-as-usual group had more total life events than the foster care and never-institutionalized group, marginally more dependent events than the foster care group, and more independent events than the never-institutionalized group. The foster care and never-institutionalized children did not differ on their experience of any class of life events. Thus, there was a modest intervention effect of foster care on total and dependent life events, but not independent events.

**Stress sensitization and protection by foster care.** Our primary analysis examined the association between stressful life events at age 12 and externalizing problems at age 16 as a function of

**Table 1 Bivariate correlations between study variables.**

|  | 1. | 2. | 3. | 4. | 5. | 6. | 7. | *M* | *SD* |
|---|---|---|---|---|---|---|---|---|---|
| 1. Total life events (age 12) | – |  |  |  |  |  |  | 3.22 | 2.48 |
| 2. Dependent life events (age 12) | 0.75*** | – |  |  |  |  |  | 1.19 | 1.21 |
| 3. Independent life events (age 12) | 0.87*** | 0.36*** | – |  |  |  |  | 1.94 | 1.69 |
| 4. Externalizing problems (age 16) | 0.26** | 0.14 | 0.29** | – |  |  |  | 0.00b | 0.97 |
| 5. Externalizing problems (age 12) | 0.20* | 0.16† | 0.19* | 0.35*** | – |  |  | 0.00b | 0.99 |
| 6. Sex (male) | 0.02 | 0.04 | 0.03 | 0.18* | 0.27** | – |  | – | – |
| 7. Age when entered institution | 0.15 | 0.15 | 0.11 | −0.01 | −0.09 | −0.13 | – | 2.87 | 4.07 |
| 8. Age at placement into foster carea | 0.16 | 0.16 | 0.11 | 0.02 | 0.10 | 0.37** | 0.26† | 23.40 | 6.71 |

***$p < 0.001$. **$p < 0.01$. *$p < 0.05$. †$p < 0.10$
aCorrelation is only within the foster care group
bThese are standardized factor scores with means of zero
*Note.* Variables below the solid line are covariates or secondary variables of interest

**Table 2 Stress sensitization effect on externalizing problems at age 16.**

| Type of life events | Institutional rearing group | | | | | |
|---|---|---|---|---|---|---|
|  | **Ever-institutionalized** | | | **Never-institutionalized** | | |
|  | *B* | 95% CI | β | *B* | 95% CI | β |
| Total | 0.11* | [0.01,0.20] | 0.21 | 0.07 | [−0.04,0.18] | 0.24 |
| Dependent | 0.06 | [−0.11,0.24] | 0.07 | 0.13 | [−0.09,0.35] | 0.22 |
| Independent | 0.19** | [0.06,0.32] | 0.28 | 0.08 | [−0.07,0.23] | 0.19 |

**$p < 0.01$. *$p < 0.05$
*Note.* Results from regression analysis, where all coefficients are adjusted for gender and externalizing problems at age 12

**Table 3 Stress-buffering effect on externalizing problems at age 16.**

| Type of life events | Intent-to-treat intervention group | | | | | |
|---|---|---|---|---|---|---|
|  | **Care-as-usual** | | | **Foster care** | | |
|  | *B* | 95% CI | β | *B* | 95% CI | β |
| Total | 0.18* | [0.01,0.36] | 0.30 | 0.02 | [−0.09,0.12] | 0.04 |
| Dependent | 0.09 | [−.21,0.40] | 0.09 | −0.02 | [−0.22,0.18] | −0.03 |
| Independent | 0.32** | [0.10,0.54] | 0.41 | 0.05 | [−0.10,0.20] | 0.09 |

**$p < 0.01$. *$p < 0.05$
*Note.* Results from regression analysis, where all coefficients are adjusted for gender and externalizing problems at age 12

group. In this analysis, we controlled for child sex and prior levels of externalizing problems at age 12. Statistics for this analysis are reported in Tables 2 and 3. First, there was a significant positive association between the number of total stressful life events at age 12 and externalizing problems at age 16 among ever-institutionalized children, but not never-institutionalized children (Table 2).

Second, in the intent-to-treat analysis, there was a significant positive association between the number of total stressful life events and externalizing problems among care-as-usual children, but not foster care children (Table 3). Thus, more stressful life events at age 12 predicted higher externalizing problems at age 16 among children with prolonged institutional rearing, but not those assigned to early foster care, indicating an intervention effect (see Fig. 1). This pattern of results was not simply driven by the care-as-usual group reporting more extreme levels of life events (e.g., 7+ events), as the care-as-usual and foster care group did not differ on their experience of any specific number of events (see Table 4). In other words, although care-as-usual children reported more total events across the entire scale than foster care

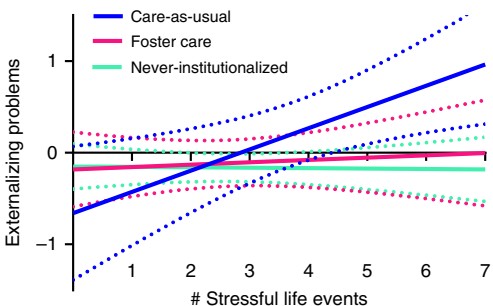

**Fig. 1 Stress sensitization model.** Association between number of total stressful life events at age 12 and externalizing problems at age 16 among care-as-usual, foster care, and never-institutionalized children. The dependent variable is a standardized factor score with a sample mean of zero. Dotted lines are 95% confidence bands. The only association that was significant was for care-as-usual children (exact statistics presented in Tables 2 and 3).

**Table 4 Descriptive statistics and sample demographics for study groups at age 12.**

|  | Never-institutionalized ($n = 44$) | Care-as-usual ($n = 48$) | Foster care ($n = 50$) |
|---|---|---|---|
| *Sex* | | | |
| Male | 45.5% (20/44) | 56.3% (27/48) | 48.0% (24/50) |
| Female | 54.5% (24/44) | 43.8% (21/48) | 52.0% (26/50) |
| *Ethnicity* | | | |
| Romanian | 94.9% (37/39) | 48.9% (23/47) | 58.3% (28/48) |
| Rroma (gypsy) | 5.1% (2/39) | 40.4% (19/47) | 29.2% (14/48) |
| Unknown | 0% (0/39) | 10.6% (5/47) | 12.5% (6/48) |
| *# Stressful life events* | | | |
| 0 | 11.4% (5/44) | 2.1% (1/48) | 12.0% (6/50) |
| 1 | 22.7% (10/44) | 10.4% (5/48) | 24.0% (12/50) |
| 2 | 20.5% (9/44) | 20.8% (10/48) | 18.0% (9/50) |
| 3 | 15.9% (7/44) | 20.8% (10/48) | 20.0% (10/50) |
| 4 | 6.8% (3/44) | 10.4% (5/48) | 4.0% (2/50) |
| 5 | 6.8% (3/44) | 10.4% (5/48) | 8.0% (4/50) |
| 6 | 11.4% (5/44) | 10.4% (5/48) | 2.0% (1/50) |
| 7+ | 4.5% (2/44) | 14.6% (7/48) | 12.0% (6/50) |

*Note.* Differences between groups on each discrete number of life events were assessed using a *z*-test that contrasts column proportions. There were no group differences for any discrete number of life events

children, the two groups did not differ statistically on whether they experienced any single number of events (e.g., 0 events, 1 event, 2 events, etc.).

**Dependent versus independent events.** To determine whether specific life events drove the effects documented above, we split the life events measure into dependent and independent life events (see Supplementary Materials for classification). The distribution on these two variables was similar: dependent (0 = 35.2%, 1 = 31.7%, 2 = 18.3%, 3 = 10.6%, 4+ = 4.2%); independent (0 = 21.1%, 1 = 26.1%, 2 = 22.5%, 3 = 12.7%, 4 = 9.2%, 5+ = 8.5%). The number of dependent life events at age 12 was not associated with externalizing problems at age 16 among either never- or ever-institutionalized children (Table 2), and was not associated with externalizing problems among care-as-usual or foster care children in the intent-to-treat analysis (Table 3). In contrast, there was a significant positive association between the number of independent life events and externalizing problems among ever-institutionalized children, but not never-institutionalized children. In the intent-to-treat analysis, there was a significant positive association between the number of independent events and externalizing problems among care-as-usual, but not foster care, children. Thus, more independent life events predicted higher externalizing problems among those with prolonged institutional care, but not those assigned to foster care or never-institutionalized children. (We tested an alternative model in which childhood neglect was proposed to increase risk of externalizing problems in adolescence through stressful life events at age 12. This stress generation hypothesis is presented in the Supplementary Materials. Briefly, this model was less suggestive than the stress sensitization model, with weak evidence that the effect of early neglect on externalizing problems operated indirectly through life events at age 12 (whether assessed as total, dependent, or independent events). Thus, there was more evidence for stress sensitization than stress generation in the current study.).

**Planned sensitivity analysis.** We conducted a sensitivity analysis to determine whether the group differences on stress sensitization documented above were, in fact, explained by time spent in institutions. For this analysis, only ever-institutionalized children were included in the analysis, and the percentage of one's life spent in institutions up to age 16 was used as a continuous moderator variable. Regression analysis showed that there was a marginally significant interaction between total life events and percent time spent in institutions on externalizing problems at age 16, $B = 0.003$, 95% CI [0.00, 0.01], $\beta = 0.18$, $p = 0.08$, and a significant interaction between the number of independent events and percent time in institutions on externalizing problems, $B = 0.006$, 95% CI [0.001, 0.01], $\beta = 0.25$, $p = 0.01$. The pattern of these interactions is presented in Supplementary Fig. 2, and is consistent with Fig. 1 in showing that more stressful life events at age 12 predicted higher externalizing problems at age 16, and this effect was stronger as time spent in institutions increased.

**Post hoc specificity analysis.** Finally, we examined whether stress sensitization also manifested for internalizing problems and a general psychopathology factor that captured shared variance across all symptom domains derived from the same previous investigation as the externalizing problems factor. There was no evidence of stress sensitization for these outcomes, suggesting that this effect is specific to externalizing problems in this sample of children.

**Discussion**

In the current study we showed that profound early neglect as a function of institutional rearing sensitizes children to the effects of later stressful life events on externalizing problems in adolescence. Specifically, we observed that more stressful life events at age 12 were associated with higher levels of externalizing problems at age 16 only among children who were not randomly assigned out of institutional care early in life. In contrast, children who were either never institutionalized or who were randomly assigned to high-quality foster care did not show increased externalizing problems as a function of more stressful life events. On aggregate, these results suggest that a history of prolonged childhood neglect increases vulnerability to the effects of later stressful events proximal to externalizing problems. Perhaps more compelling, we provide experimental evidence that early social fortification afforded by family care buffers the sensitizing effects of childhood neglect to later stressful life events on psychopathology. These effects held after controlling for child sex, covariance with other mental health problems, and prior levels of externalizing problems, strongly supporting the stress-sensitizing effect of early neglect, and the stress-buffering effect of enriched caregiving, on externalizing problems in adolescence.

At a more fine-grained level of analysis, the stress-sensitizing and stress-buffering effects reported above were most pronounced for stressful life events over which individuals had little control (i.e., independent life events). This lack of control over the environment during a formative period of individuation may be particularly stress-inducing among adolescents with histories of adversity. Similar effects have been observed for internalizing difficulties such as depression during adolescence[15,17]. Together, these findings highlight the possibility that independent stressful events may have a considerable impact on psychological development when they occur during periods of marked physiological reorganization[10]. Our results align with a longstanding literature on diathesis-stress, which underscores how susceptibility to mental health difficulties is amplified by later-occurring stressors[26]. Stress sensitization is a specific example of diathesis-stress wherein early adversity represents a general diathesis, and later or prolonged exposure to stress is required to trigger psychopathology. These results suggest that independent life events during preadolescence may be an especially potent stressor that aggravates underlying vulnerabilities set forth by early adversity. While these results are consistent with the notion that adolescence may be a sensitive period in development, they do not confirm such claims. Future studies that directly contrast the effects of stress sensitization and buffering between children, adolescents, and adults are required to determine at what developmental stage these effects are most operative. Moreover, determining whether interpersonal stressors exert a stronger effect than non-social stressors on behavior over the transition to adolescence will improve our understanding of the conditions under which stress increases susceptibility to psychopathology among those with histories of adversity.

The precise mechanisms that account for the stress-sensitizing and stress-buffering effects reported in the current study are unknown. One candidate pathway involves dysfunction of the autonomic nervous system and neuroendocrine system (HPA-axis). Lower cortisol levels and blunted reactivity of the HPA-axis to stress-inducing stimuli has been linked to higher levels of externalizing problems[26]. Furthermore, there is some evidence that hyporeactivity of stress systems may associate more strongly with externalizing problems, while hyperreactivity may relate more strongly to internalizing problems[27–30]. Within the BEIP, children with prolonged institutional rearing show blunted HPA-axis and sympathetic nervous system reactivity to laboratory stressors in preadolescence. In contrast, stress reactivity of children in the foster care group approximates that of the never-institutionalized group, suggesting remediation of the adaptive stress response as a function of social enrichment, especially if this occurs within the first two years of life[31]. Moreover, cortisol hyporeactivity has been shown to mediate the association between child maltreatment and later externalizing problems in other studies of post-institutionalized children[26], and recent evidence suggests that early and later stress interact in predicting flatter diurnal cortisol slopes[32]. Together, these findings raise the possibility that blunting of stress systems may be a physiological mechanism of stress sensitization, which in turn predisposes youth to later externalizing problems.

While the primary goal of this study was to examine whether the association between stressful life events and externalizing problems in adolescence varied as a function of children's exposure to early institutional deprivation, it is worth noting that there was a modest intervention effect on the experience of stressful life events, in particular dependent life events. One potential factor to explain why the foster care children reported fewer dependent events than the care-as-usual children is differences in caregiving quality experienced by these groups. Supplementary analyses showed that not only do the foster care

children experience higher quality caregiving than the care-as-usual children at ages 12 and 16 years, but the association between caregiving quality and stressful life events appears stronger for the foster care group (see Supplementary Results). This suggests that the foster care children may be more responsive to the higher quality care they receive, and this higher quality care may be protective against later stressful life events. In turn, this may help to explain the lower levels of externalizing problems observed between these groups at age 16[5]. Future studies can help to determine whether the putative buffering effect of higher caregiving quality operates through its influence on stress system development and emerging self-regulatory abilities in childhood and adolescence.

The finding that early placement into foster care attenuates the association between stressful life events and externalizing problems in adolescence is consistent with recent work showing lower externalizing problems among foster care compared with care-as-usual children in preadolescence[2]. Interestingly, the protective effect of foster care on externalizing problems was not observed at 54 months in BEIP[4]. Consequently, the results of the current study suggest that one mechanism by which foster care may protect institutionally-reared children against the emergence of externalizing problems later in childhood is by buffering them against stress during periods of significant neural reorganization. It is also noteworthy that children assigned to early foster care show remediation in global white matter volume[33] and white matter tract integrity[34], as well as improved accuracy, quicker neural processing, and enhanced error detection on measures of inhibitory control[35]. Together, these results suggest that social and interpersonal enrichment afforded by foster care may be associated with remediation in neural structure and function that equips children with the cognitive and self-regulatory skills to more effectively manage later stress, thus reducing the risk of externalizing problems during adolescence.

The current study should be considered in light of several limitations. First, the relatively small sample may have limited our power to detect certain effects, especially intervention-timing effects, which did not manifest for externalizing problems. Second, our measurement of externalizing problems was derived from teacher and caregiver ratings. While combining ratings from both respondents reduces the potential for rater bias, replication using diagnostic interviews is warranted. Moreover, although the RCT design takes account of some baseline differences between the care-as-usual and foster care groups, other factors that were not accounted for (e.g., socioeconomic status, subsequent placement) may explain additional variation in life events, externalizing problems, or both. Third, there was some attrition over time, which is perhaps unsurprising given the 16-year follow-up in a high-risk sample. Our ability to detect the documented effects even after controlling for covariance with other domains of psychopathology, child sex, and prior levels of externalizing problems within this sample improves confidence in the robustness of effects. Fourth, as children were not randomly assigned to high versus low levels of life events, we cannot definitely conclude that stressful life events are causally linked to higher externalizing problems. However, Supplementary Fig. 3 showed that more stressful life events at age 12 predicted higher externalizing problems at age 16, while externalizing problems at age 12 only weakly predicted life events at age 16. This effect was independent of children's history of institutionalization. Thus, the relation between life events and problematic behavior appears to be directional in this sample, and cannot simply be attributed to differences in life events across groups. Finally, our sample comprised a fairly unique group of severely neglected children, and it is not clear whether the current results generalize to less profound forms of neglect or other forms of adversity. Future

research that compares institutionalized youth with those exposed to other forms of adversity will help to characterize the shared and distinct mechanisms underpinning susceptibility to psychopathology in adolescence and beyond.

The findings reported in the current study have implications for the well-being of children around the world who have experienced institutional care early in life. Caregivers and professionals providing services for these children should be aware of the harmful effects that additional stress may have on emerging psychopathology in adolescence, particularly if such stressors are beyond the child's control (what we refer to as independent life events). Moreover, adolescence is an age at which children's social world expands beyond the family and where they become increasingly independent. Because children with a history of institutional care may be ill-equipped to deal with emergent stressors during adolescence, there is a critical need to ensure effective adaptation and coping during this period. Preventative

counseling and therapeutic services should be made available by psychologists and social workers to help children proactively identify potential stressors and generate effective solutions and responses to stress that accompany both anticipated and unanticipated life events.

## Methods

**Participants**. Participants were children from the Bucharest Early Intervention Project (BEIP; clinicaltrials.gov identifier: NCT00747396), a longitudinal study of children with histories of severe psychosocial deprivation who were reared in institutions in Bucharest, Romania. BEIP is the only RCT of foster care as an alternative to institutional care. The study began in April 2001, and the most recent (age 16) follow-up was completed in October 2018. At an average age of 22 months (range = 6–31), children were recruited from six institutions in Bucharest, Romania. At baseline, the children were assessed by a pediatrician experienced with young children raised in institutions and excluded based on the presence of genetic syndromes, fetal alcohol syndrome, and micro- or macrocephaly. A total of 136 children met the inclusion criteria and were subsequently randomized to remain in institutional care (care-as-usual group) or to leave the institutions and enter the

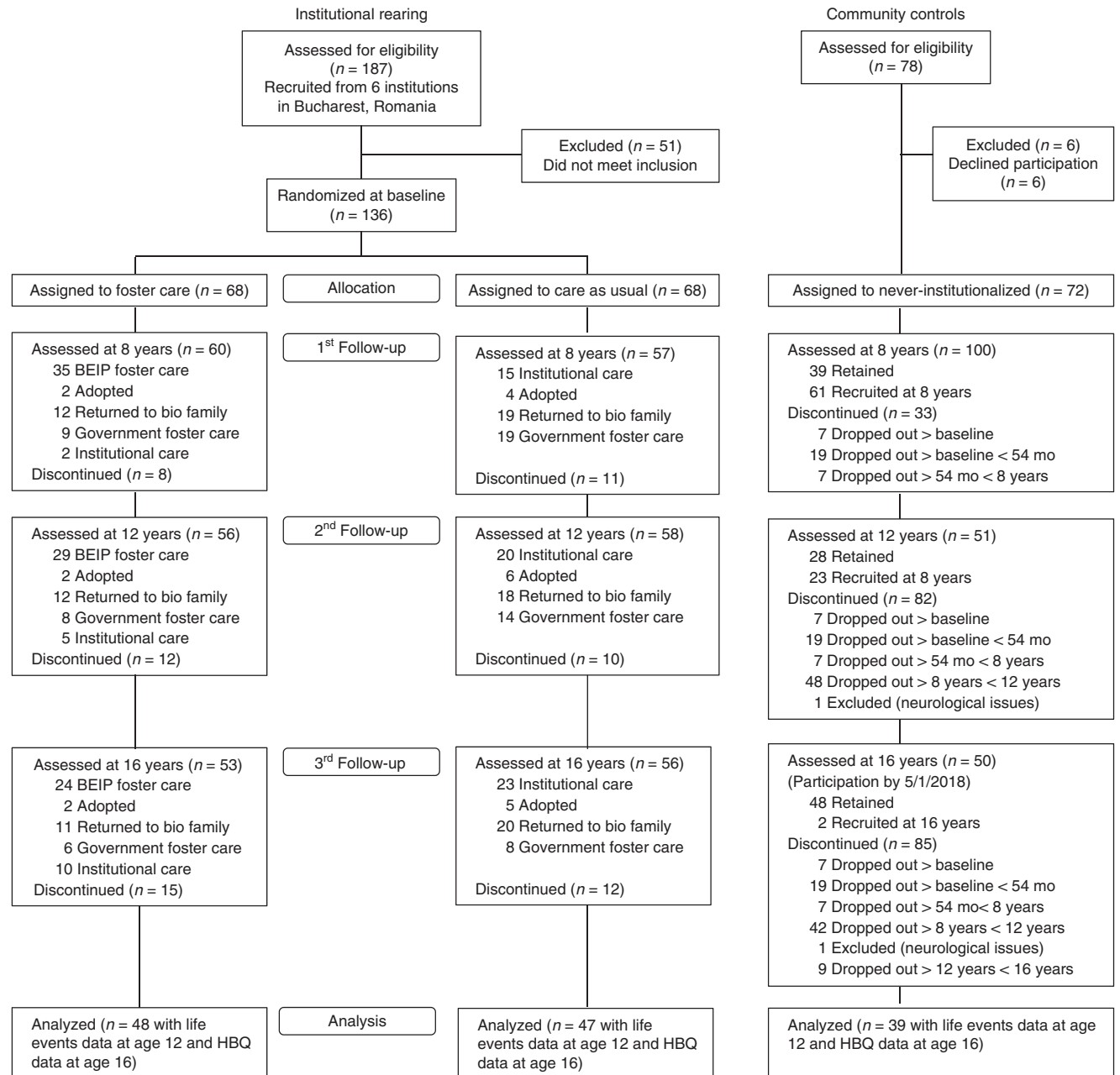

**Fig. 2 CONSORT flow diagram.** Group assignment and follow-up measurement in the randomized controlled trial.

care of one of the foster families identified and trained by study investigators (foster care group) by drawing names from a hat. These two groups together comprise the ever-institutionalized group. An age-matched sample of 72 never-institutionalized children reared in their biological families was recruited from pediatric clinics in Bucharest to serve as a comparison group. BEIP adopted a non-interference policy throughout the duration of the study. Thus, although most care-as-usual children remained in institutional care through age 5, many were removed from institutional care at some point and, by ages 12 and 16 years, more than half were living in some kind of family placement (see Fig. 2, CONSORT diagram).

Signed consent was obtained from each child's legal custodian. Written assent was obtained from each child beginning at age 12, unless the child had an intellectual disability, in which case they gave verbal assent. Never-institutionalized children also gave written assent beginning at age 12 and their legal guardians gave signed consent.

BEIP was originally initiated at the request of the Secretary of State for Child Protection in Romania. All study procedures were approved by the local Commissions on Child Protection in Bucharest, the Romanian Ministry of Health, and the institutional review boards of the three principal investigators (CAN, NAF, CHZ). We and others have discussed ethical considerations of this study, which can be found elsewhere[36].

While the formal RCT terminated at 54 months of age, extensive follow-ups on the children were conducted at 8, 12, and 16 years. We collected data on self-reported stressful life events from 142 participants (care-as-usual = 48; foster care = 50, never-institutionalized = 44) in preadolescence (age 12). At age 16, caregivers and teachers reported on psychopathology for 149 participants (care-as-usual = 49; foster care = 52, never-institutionalized = 48). There was overlap in measurement for 134 children (care-as-usual = 47; foster care = 48, never-institutionalized = 39); thus, our final sample was $N = 134$. Table 4 provides characteristics of the sample by group. No significant differences were found between the care-as-usual, foster care, and never-institutionalized groups in terms of age or sex distribution.

**Stressful life events**. At age 12, children self-reported on the presence/absence of up to 30 life events that happened to them or members of their family over the past 12 months. The measure is a modified version of Coddington's Child Life Events Scale[37] that was adapted for use in this sample. Example items included: "you failed a grade in school or got bad grades"; "you and your boyfriend/girlfriend had a big fight or broke up"; "your family's house or car was broken into or robbed"; and "you had a serious accident or illness and were in the hospital." The variable was significantly right-skewed, with only 8.5% of the sample ($n = 12$) reporting between 8 and 12 (the maximum) events. Thus, we re-scaled the item to reduce this skew by combining children reporting 7 or more life events. The distribution on this variable was as follows: 0 = 8.5%; 1 = 19.0%; 2 = 19.7%; 3 = 19.0%; 4 = 7.0; 5 = 8.5; 6 = 7.7; 7+ = 10.6%. This variable was then split into dependent and independent life events as noted above. The full scale can be found in the Supplementary Materials, including differentiation of dependent and independent events.

**Externalizing problems**. At age 16, the MacArthur Health and Behavior Questionnaire was administered to the children's caregivers ($N = 146$) and/or teachers ($N = 81$) to assess symptoms in several domains of psychopathology: depression, overanxious, social anxiety, oppositional defiant, conduct problems, overt aggression, relational aggression, and ADHD (see the following for details: https://macarthurhbq.wordpress.com/). Caregiver and teacher ratings were significantly correlated for all domains of psychopathology ($r = 0.24–0.56$, all $p$'s <0.05), and were thus combined into composite scores to reduce rater bias. We used a previously-derived saved factor score for externalizing problems that was estimated from a latent bifactor model[38]. In this model, a general psychopathology factor accounted for the shared variance across all psychopathology domains, and the externalizing factor captured the overlap between externalizing dimensions (oppositional defiant, conduct problems, overt aggression, relational aggression, and ADHD) after accounting for the variance in the general factor. Thus, the primary outcome was externalizing problems at age 16. Importantly, we also controlled for externalizing problems at age 12, concurrent with our measurement of stressful life events. We previously established measurement invariance on these latent factors over time[38].

**Statistical analysis**. The analyses were conducted in SPSS version 21 and Mplus version 7. Two sets of analyses were carried out for the stress sensitization model. The first analysis ($N = 134$) compared never-institutionalized with ever-institutionalized children, and thus examined differences based on one's history of institutional rearing. The second analysis ($N = 95$) used an intent-to-treat approach to examine whether assignment to foster care was associated with less problematic outcomes compared with care-as-usual (i.e., prolonged institutional care). The intent-to-treat analysis compared children on the basis of their initial placements, and allows for group differences to be interpreted as the causal effect of randomization into foster care on outcomes.

The association between total stressful life events at age 12 and externalizing problems at age 16 was examined using multi-group linear regression models that controlled for sex and prior levels of externalizing problems at age 12. These were therefore auto-regressive models, which improve inferences about the direction of effects by testing how life events are associated with change in externalizing problems over time[39]. The analyses were then repeated for dependent and independent life events separately in order to determine whether one class of event was driving the pattern of results.

Finally, we conducted a sensitivity analysis to examine whether, in fact, prolonged institutional rearing was associated with increased externalizing problems as a function of more life events. In this moderation analysis, the duration of institutionalization among ever-institutionalized children was used as a continuous moderator variable in interaction with the number of life events in predicting externalizing problems.

**Reporting summary**. Further information on research design is available in the Nature Research Reporting Summary linked to this article.

## Data availability
These data are available from the authors on request. The source data underlying Fig. 1 and Supplementary Figs. 2–4 are provided as a Source Data file. A reporting summary for this Article is available as a Supplementary Information file.

## Code availability
The syntax used to analyze the data in SPSS are available from the authors on request.

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

## Acknowledgements
We are grateful to all the children and caregivers who participated in the Bucharest Early Intervention Project and to our dedicated lab staff in Romania. This study was supported by the John D. and Catherine T. MacArthur Foundation, the Binder Family Foundation, NIMH (R01MH091363) to Charles Nelson, and a Banting Fellowship to Mark Wade. The funding sources had no role in the design of the study; collection, management, analysis, and interpretation of the data; preparation, review, or approval of the manuscript; or decision to submit the manuscript for publication.

## Author contributions
N.A.F., C.H.Z., and C.A.N. designed and carried out the study. M.W. processed and analyzed the data. F.T. was part of the team in Romania responsible for data collection and participant recruitment. L.E.C. offered expertise in the context of institutionalization in Romania and impact of BEIP on policy and programming. M.W., N.A.F., C.H.Z., and C.A.N. wrote the paper; and F.T. and L.E.C. reviewed the final draft.

## Competing interests
Dr. Nelson has received grant support from NIH, the Jacobs Foundation, the John D. and Catherine T. MacArthur Foundation, the James S. McDonnel Foundation, the Binder Family Foundation, the Lumos Foundation, the Bill and Melinda Gates Foundation, and Harvard University. He has received royalties from MIT and Harvard University Press. He has received honoraria for lectures to professional audiences, and legal consulting fees. Dr. Zeanah has received grant support from NIMH, the Palix Foundation, the Irving Harris Foundation, the Substance Abuse and Mental Health Services Administration, and the Lumos Foundation. He has received royalties from Guilford Press and Harvard University Press. He has received honoraria for lectures to professional audiences. Dr. Fox receives grant support from NIMH, NICHD, NSF, the NIH ECHO consortium, the Russell Sage Foundation, and the Lumos Foundation. He has received royalties from Guilford Press and Harvard University Press. He has received honoraria for lectures to professional audiences. Drs. Wade, Tibu, and Ciolan declare no competing interests.
