## [Peer Review File · Nature Communications]

Reviewers' Comments:

Reviewer #1:

Remarks to the Author:

Wade et al provide data on the follow-up of an RCT with Romanian children raised in institutions. This is a unique cohort and the imposition of the RCT has provided unique evidence for causal influences of early life deprivation and the reversibility (or lack of) deprivation effects. In the current paper the authors show that institutional rearing sensitizes children to the effects of later stressful life events on externalizing problems in adolescence. The core finding is that stressful life events predicted increased externalizing problems in the institutionalized children, but not in those who move to foster care. The inclusion of a never-institutionalized group provides an invaluable comparison group. Although the sample size is modest (as the authors acknowledge) the data are compelling. I note in particular the β -values for the association between independent stressful events and externalizing outcomes for the Care-as-Usual and Foster-care groups (also see Figure 2). An additional strength was the association between total stressful life events at age 12 and EXT at age 16 was examined using multi-group linear regression models that controlled for prior levels of EXT at age (Lines 205-206).

Comments

As a brief note on presentation, please limit the use of abbreviations – or eliminate them entirely. The paper is targeting a broad audience and the use of such esoteric abbreviations diminishes the clarity of an otherwise well-written paper.

It is important not to over-interpret the findings. The authors (e.g., abstract) claim to have examined “externalizing psychopathology”. They have not. They have obtained caregiver and teacher reports. These are not validated diagnostic screens. I think a more conservative term would be “externalizing problems” – this change would not diminish the impact of the paper. I realize that the earlier paper published in JAMA Psychiatry obviously did not impose this limitation. But parent and teacher reports of internalizing and externalizing problems do not align nearly as well to overt diagnosis as is commonly assumed. One could argue this issue has much to do with diagnostic systems – but the point is the actual outcome measure was reported behavioural problems.

Lines 62-65: The authors correctly state “Despite the significant association between stressful life events and psychopathology in adolescence, not all individuals who experience such stressors will develop psychiatric disorders. One hypothesis to explain this inter-individual variability is that risk for psychopathology is heightened among those who have also experienced childhood adversity.” This is a critical point for the paper. I am surprised that the authors have not expanded on this point in the Discussion. The data bear on stress-diathesis models.

This point and the importance of the predictive impact of dependent vs independent stressful life events could be elaborated in the Discussion.

The Discussion should also be more focused on the actual findings. Musings on the potential relation to HPA function are less relevant. First, levels of cortisol show little specificity to forms of psychopathology and the affective – behavioral effects of acute HPA activation are more closely linked to CRF. Likewise, the amygdala – PFC imaging results are potentially interesting, but require a more specific coupling to the current results. Also, amygdala – PFC connectivity commonly associates more strongly with internalizing issues, such as symptoms of depression and anxiety. I would argue for reducing these sections considerably in favor of material that bears more closely on the core findings

(see comment above).

Lines 186-187: Please provide the range of the actual correlations between caregivers and teachers.

Lines 181 -. The authors should reference their previous paper (JAMA Psychiatry) that provides a justification for the exclusive focus on externalizing problems.

Lines 226-229: How would the authors account for the finding that "...CAUG had more total life events than FCG and NIG... Thus, there was a modest intervention effect of foster care on total and dependent life events, but not independent events."

Reviewer #2:

Remarks to the Author:

This study used data from the Bucharest Early Intervention Project to examine whether early neglect increased vulnerability to the effects of later stressful life events with regard to externalizing symptoms during adolescence. The authors report that youth who experienced prolonged institutional rearing (i.e., care as usual group; CAUG) showed an association between more stressful life events in preadolescence and higher externalizing symptoms in adolescence, whereas youth assigned to foster care and never-institutionalized children did not show this effect. Strengths of the research include the unique study design and important question. Despite its strengths, it is difficult to evaluate the results as evidence for stress sensitization, the primary focus of the study.

Because the number of total stressful life events differed by group (significantly higher in the CAUG, the same group showing the association between life events and later externalizing symptoms; currently detailed in the Supplement), the evidence for stress sensitization is not clear. It is not possible to dissociate this heightened exposure to stressful life events from the early experience of prolonged institutional care. The authors appear to attempt to overcome this issue by pointing out that the groups do not differ on any specific number of life events, despite differing on the total number of life events. However, this does not address the problem.

The sole focus on externalizing symptoms was unexpected. Could the authors test the same relationship for internalizing symptoms? If specific to externalizing, that information would be important to add.

Pg. 9 typo: "McArthur" should be "MacArthur"

Supplement typo: "depdendent" should be "dependent"

Reviewer #3:

Remarks to the Author:

Overview: The goal of this paper was to examine whether severe early neglect among children reared in institutions increases vulnerability to the effects of later stressful life events on externalizing psychopathology in adolescence, and whether social enrichment in the form of high-quality foster care buffers this risk. There were not main effects of the intervention itself on externalizing problems in adolescence, but these effects were apparent among a sub-group who had not been adopted into foster care and had experienced more stressful life events in preadolescence. The paper is on a fascinating topic, but could use some additional clarity on the following issues.

Larger Concern 1: Since there are only 7 of the CAUG kids who scored 7+ on the Stressful Life Events Scale, and 5 in the next category, and 5 in the next, is it possible that outliers are at work, which could explain the findings presented Figure 2? Perhaps some robustness checks could be conducted to explore what is happening at the tails of the distribution of Stressful Life events scale. In fact, there are so few children in each category, it seems like the best thing to do would be to condense to 3 buckets so that each bucket has 10-15 in it.

Larger Concern 2: The question of causality is somewhat confusing in this paper. Are stressful life events hypothesized to lead to externalizing behavior or the other way around? If stressful life events is the "outcome" and EXT is the "mediator", you are assuming that one leads to the other. Wouldn't it be equally plausible that stressful life events could lead to externalizing behavior and acting out? Some sort of theoretical model or textual framing would help here in elucidating the ways that the authors are imaging their conceptual framework. I can see you that you have a footnote describing this alternative hypothesis, but I wonder if it wouldn't help to discuss more clearly in the intro how you imagine the direction of causality.

Larger Concern 3: Where is the list of other covariates that would have been included? E.g. household wealth, family educational status, size of household, demographic structure of household, type of school child attends, whether parents are married, parental mental health, etc. There are so many variables that would affect both life events and externalizing behavior and it is hard to ignore them here.

Other Concerns

Methods: Question about missingness – how are the 134 children included in the study similar/different from the total of 149. The discussion raises this issue as a limitation, but then says it is not an issue because there were still significant findings. The issue, though, is what type of kid you might have lost, and how that would have changed your inferences based on your sample. You could consider IPW (inverse probability weighting) as a sensitivity analysis to make sure that there are no issues with your existing sample being fundamentally different from the 10% who are missing.

Methods/Results: Stressful Life Events (text & table 1): Were tests of difference among groups examined, and if so what tests were used? The footnote of the table suggested that there were no significant differences. In the text, the authors describe that the items were re-scaled to reduce skew. How were tests of difference conducted with this variable?

Methods/Results: There is something a little confusing about Table 1, since it is double counting two sets of children. I wonder if it wouldn't be simpler to present three columns – NIG, CAUG, FCG. It also raises the question of multiple hypothesis testing if you are looking at all of these different comparisons across group. Have you adjusted for multiple hypothesis testing?

Methods/Results: Externalizing Psychopathology. As presented, this variable sounds like another outcome measure, with the hypothesis that institutionalization would increase externalizing. But the description mentions that "this factor is used as a covariate".

Methods/Results: Dependent and independent life events? What is the difference? This terminology is used in the statistical analysis section but does not appear above in the description of the variable.

Methods/Results: If EXT wasn't used as an outcome measure, it is unclear how it was used in the model. How does it fit conceptually with the system you are testing?

Results: Not clear what this result means: "this effect was stronger as time spent in institutions increased" Where are these results presented?

Discussion: Do you have any explanations for why children who went into foster care were receiving "buffering them against stress during periods of significant neural reorganization." Do you know that this is true from earlier work? Or is this just speculation?

Minor comments:

Figure 2 could use description of acronyms

Might be helpful to have website link for "MacArthur Foundation Research Network for Psychopathology and Development"

Reviewers' comments:

Reviewer #1 (Remarks to the Author):

Wade et al provide data on the follow-up of an RCT with Romanian children raised in institutions. This is a unique cohort and the imposition of the RCT has provided unique evidence for causal influences of early life deprivation and the reversibility (or lack of) deprivation effects. In the current paper the authors show that institutional rearing sensitizes children to the effects of later stressful life events on externalizing problems in adolescence. The core finding is that stressful life events predicted increased externalizing problems in the institutionalized children, but not in those move to foster care. The inclusion of a never-institutionalized group provides an invaluable comparison group. Although the sample size is modest (as the authors acknowledge) the data are compelling. I note in particular the β -values for the association between independent stressful events and externalizing outcomes for the Care-as-Usual and Foster-care groups (also see Figure 2). An additional strength was the association between total stressful life events at age 12 and EXT at age 16 was examined using multi-group linear regression models that controlled for prior levels of EXT at age 12 (Lines 205-206).

Comments:

As a brief note on presentation, please limit the use of abbreviations – or eliminate them entirely. The paper is targeting a broad audience and the use of such esoteric abbreviations diminishes the clarity of an otherwise well-written paper.

***RESPONSE:** Thank you for raising this. We have now removed most abbreviations specific to this study and replaced them with full construct/variable names.*

It is important not to over-interpret the findings. The authors (e.g., abstract) claim to have examined “externalizing psychopathology”. They have not. They have obtained caregiver and teacher reports. These are not validated diagnostic screens. I think a more conservative term would be “externalizing problems” – this change would not diminish the impact of the paper. I realize that the earlier paper published in JAMA Psychiatry obviously did not impose this limitation. But parent and teacher reports of internalizing and externalizing problems do not align nearly as well to overt diagnosis as is commonly assumed. One could argue this issue has much to do with diagnostic systems – but the point is the actual outcome measure was reported behavioural problems.

***RESPONSE:** We agree with the reviewer’s comment and have changed the term “externalizing psychopathology” to “externalizing problems” throughout the paper. We still retain the term “psychopathology” when describing mental health difficulties more generally, but use “externalizing problems” when describing our study outcome specifically.*

Lines 62-65: The authors correctly state” Despite the significant association between stressful life events and psychopathology in adolescence, not all individuals who experience such stressors will develop psychiatric disorders. One hypothesis to explain this inter-individual variability is that risk for psychopathology is heightened among those who have also experienced

childhood adversity.” This is a critical point for the paper. I am surprised that the authors have not expanded on this point in the Discussion. The data bear on stress-diathesis models.

This point and the importance of the predictive impact of dependent vs independent stressful life events could be elaborated in the Discussion.

RESPONSE: *We agree this point is deserving of elaboration, and we now expand on this, including reference to the stress-diathesis literature, in the Discussion (pg. 16).*

The Discussion should also be more focused on the actual findings. Musings on the potential relation to HPA function are less relevant. First, levels of cortisol show little specificity to forms of psychopathology and the affective-behavioral effects of acute HPA activation are more closely linked to CRF. Likewise, the amygdala-PFC imaging results are potentially interesting, but require a more specific coupling to the current results. Also, amygdala-PFC connectivity commonly associates more strongly with internalizing issues, such as symptoms of depression and anxiety. I would argue for reducing these sections considerably in favor of material that bears more closely on the core findings (see comment above).

RESPONSE: *Thank you for raising these issues. We are in full agreement with the assessment made by the Reviewer on the tenuous link between cortisol and specific forms of psychopathology. However, there are two pieces of evidence that we think merit preservation of some of this section in the Discussion. First is recent evidence (published during the interval between our first submission and receipt of the reviews) pointing to a blunted diurnal cortisol response as function of the interaction between early and later adversity, providing direct support for the idea of cortisol blunting as a mechanism of stress sensitization (Young et al., 2019). Second is a sizeable literature that has emerged since original meta-analytic data (e.g., Alink et al., 2008) suggesting that blunting of the HPA-axis is more strongly linked to externalizing problems, whereas hyperreactivity may be more strongly linked to internalizing problems (e.g. Hagan et al., 2014; Hartman et al., 2013; Martinez-Torteya et al., 2016; Ruttle et al., 2011; Wadsworth et al., 2019). While not all studies show this dissociation, we find support for it in our study as well: using data from a prior study (McLaughlin et al., 2015), we observe that higher externalizing-specific problems are associated with lower HPA-axis and sympathetic nervous system (SNS) reactivity to social stress, while higher internalizing-specific problems are associated with heightened SNS (but not HPA) reactivity. We believe this emerging set of findings is important for delineating putative biological mechanisms of stress sensitization. We have thus retained but modified this section of the Discussion. We do agree that there are insufficient data linking amygdala-PFC connectivity with stress sensitization, and we have therefore removed this from the Discussion.*

Lines 186-187: Please provide the range of the actual correlations between caregivers and teachers.

RESPONSE: *We now provide this range of correlations (pg. 10).*

Lines 181 -. The authors should reference their previous paper (JAMA Psychiatry) that provides a justification for the exclusive focus on externalizing problems.

RESPONSE: We now reference this paper and add a brief justification for the explicit focus on externalizing problems (pg. 4).

Lines 226-229: How would the authors account for the finding that "...CAUG had more total life events than FCG and NIG... Thus, there was a modest intervention effect of foster care on total and dependent life events, but not independent events."

RESPONSE: Thank you for raising this important point. At a theoretical level, one feasible interpretation is that, if the foster care and care-as-usual children are going to differ, it is more likely to be on dependent versus independent events since independent events are, by their nature, more likely to occur haphazardly and in a manner that is unrelated to children's own behaviors, traits, vulnerabilities, etc. However, by this same logic we should expect the care-as-usual and never-institutionalized groups to be indistinguishable on independent life events, which is not the case. So this explanation may not be complete.

Instead, the difference between the foster care and care-as-usual children may be due to their responsiveness to caregiving experiences during (pre)adolescence, with the foster care children experiencing higher quality care and being buffered against the experience of stressful events to a greater degree than the care-as-usual children. Indeed, we provide new evidence in the Supplementary Materials that children in the foster care group experience higher-quality caregiving at age 12 and 16 years relative to the care-as-usual children ($t = 3.82, p < .001$ at age 12; $t = 3.14, p = .002$ at age 16). Moreover, using Poisson regression we see that, over and above the effects of group (foster care vs. care-as-usual), higher-quality caregiving is associated with lower total (wald $\chi^2 = 15.09, p < .001$), dependent (wald $\chi^2 = 10.03, p = .002$), and independent (wald $\chi^2 = 8.18, p = .004$) life events. So the foster care children experience higher-quality care, on average, compared to the care-as-usual children, and this higher-quality care is related to fewer stressful life events.

Finally, using Poisson regression again, we see that the effect of caregiving quality on stressful life events may be stronger for the foster care versus the care-as-usual children. Specifically, higher quality caregiving is associated with fewer total life events among the care-as-usual children (wald $\chi^2 = 4.39, p = .04$), but this effect is even stronger among the foster care children (wald $\chi^2 = 13.03, p < .001$). We see the same pattern for dependent life events (care-as-usual: wald $\chi^2 = 2.97, p = .085$; foster care: wald $\chi^2 = 8.61, p = .003$) and independent life events (care-as-usual: wald $\chi^2 = 2.99, p = .084$; foster care: wald $\chi^2 = 5.81, p = .016$). From these results, it also appears that the difference between the foster care and care-as-usual children is larger for dependent vs. independent events. Thus, not only do the foster care children experience higher quality care, but higher quality care may exert a stronger protective effect against stressful life events for the foster care children compared to the care-as-usual-children.

We have now added these results to the Supplementary Materials and provide a brief overview of them in the Discussion (pg. 17-18).

Reviewer #2 (Remarks to the Author):

This study used data from the Bucharest Early Intervention Project to examine whether early neglect increased vulnerability to the effects of later stressful life events with regard to externalizing symptoms during adolescence. The authors report that youth who experienced prolonged institutional rearing (i.e., care as usual group; CAUG) showed an association between more stressful life events in preadolescence and higher externalizing symptoms in adolescence, whereas youth assigned to foster care and never-institutionalized children did not show this effect. Strengths of the research include the unique study design and important question. Despite its strengths, it is difficult to evaluate the results as evidence for stress sensitization, the primary focus of the study.

Because the number of total stressful life events differed by group (significantly higher in the CAUG, the same group showing the association between life events and later externalizing symptoms; currently detailed in the Supplement), the evidence for stress sensitization is not clear. It is not possible to dissociate this heightened exposure to stressful life events from the early experience of prolonged institutional care. The authors appear to attempt to overcome this issue by pointing out that the groups do not differ on any specific number of life events, despite differing on the total number of life events. However, this does not address the problem.

RESPONSE: *We agree this is an important issue. There are two pieces of evidence that we think assuage this concern. First, as reported in the primary analysis, the “driver” of the total stressful life events effect is independent as opposed to dependent life events. In other words, we see the relation between stressful life events and externalizing problems for independent but not dependent events (also see figure below). The foster care and care-as-usual children do not differ on their level of independent events. So it is only among those who experienced prolonged institutional rearing that we see this effect, and it cannot simply be explained by the care-as-usual experiencing more independent events.*

The second piece of evidence that addresses this concern is an analysis that examines the interaction between stressful life events and group (care-as-usual vs. foster care) on externalizing problems, controlling for the main effects of both group and stressful life events. In this analysis, we see a pattern of results that directly mimics those in the main analysis. Specifically, we observe a modest interaction between group (care-as-usual vs. foster care) and total life events on externalizing problems ($\beta = -.28, p = .05$); a non-significant interaction involving dependent life events ($\beta = -.07, p = .65$), and a significant interaction involving independent events ($\beta = -.33, p = .01$). The pattern of these interactions is plotted below. It is clear from these plots that there is a prominent interaction for total and independent (but not dependent) life events, controlling for the main effect of group and the number of life events.

The sole focus on externalizing symptoms was unexpected. Could the authors test the same relationship for internalizing symptoms? If specific to externalizing, that information would be important to add.

RESPONSE: Our selection of externalizing-specific problems was indeed intentional, as this is the dimension of psychopathology that appears both most affected by institutional rearing and most amenable to the foster care intervention (Humphreys et al., 2015; Wade et al., 2018). It is also the dimension of psychopathology whereby stress system functioning most strongly mediates the effects of early adversity, including blunting of the HPA-axis and sympathetic nervous system (Busso et al., 2017; Koss et al., 2016). Using bifactor modeling (the same approach used in the current study), other studies have shown that chronic stress exposure is more strongly linked to externalizing- as opposed to internalizing-specific problems (Snyder et al., 2017). We now elaborate on this in the Introduction (pg. 4).

Having said that, we took the Reviewer's advice and examined the stress sensitization effect for internalizing-specific problems. The interaction with group (foster care vs. care-as-usual) was not significant for total ($\beta = -.17, p = .28$), dependent ($\beta = -.11, p = .48$), or independent ($\beta = -.11, p = .45$) life events. Thus, this effect appears specific to externalizing problems. We now state this on page 14 in the "Post-hoc Specificity Analysis" section.

Pg. 9 typo: “McArthur” should be “MacArthur”
Supplement typo: “depdendent” should be “dependent”

RESPONSE: *Thank you for pointing this out. We have made the corrections.*

Reviewer #3 (Remarks to the Author):

Overview: The goal of this paper was to examine whether severe early neglect among children reared in institutions increases vulnerability to the effects of later stressful life events on externalizing psychopathology in adolescence, and whether social enrichment in the form of high-quality foster care buffers this risk. There were not main effects of the intervention itself on externalizing problems in adolescence, but these effects were apparent among a sub-group who had not been adopted into foster care and had experienced more stressful life events in preadolescence. The paper is on a fascinating topic, but could use some additional clarity on the following issues.

Larger Concern 1: Since there are only 7 of the CAUG kids who scored 7+ on the Stressful Life Events Scale, and 5 in the next category, and 5 in the next, is it possible that outliers are at work, which could explain the findings presented Figure 2? Perhaps some robustness checks could be conducted to explore what is happening at the tails of the distribution of Stressful Life events scale. In fact, there are so few children in each category, it seems like the best thing to do would be to condense to 3 buckets so that each bucket has 10-15 in it.

RESPONSE: *Thank you for raising this. We agree that there is an inherent risk of outliers driving the analysis when dealing with a variable that has few individuals in the tail of the distribution. This, of course, is why we truncated the variable to be 0-7+ life events when in reality some children reported up to 12 life events. However, it is still plausible that the distribution of this variable is biasing the results. Thus, we took the Reviewer’s advice and re-scaled the variable to be 0-4+ events. We chose 4 as the maximum category because this yielded a more evenly distributed number of children across the number of life events (i.e., $N_{0events} = 12$, $N_{1event} = 27$, $N_{2events} = 28$, $N_{3events} = 27$, $N_{4events} = 48$). We re-analyzed the models with this new scaling, and the results were unchanged: There was an increase in externalizing problems as the number of total life events increased for the CAUG ($\beta = .28$, $p = .05$), but not FCG ($\beta = .01$, $p = .93$) or NIG ($\beta = .19$, $p = .28$). As in the primary analysis, this effect was not observed for dependent events, and was instead driven by independent events, with an increase in externalizing problems for CAUG ($\beta = .45$, $p = .002$), but not FCG ($\beta = .10$, $p = .47$) or NIG ($\beta = .22$, $p = .20$). Thus, we do not believe the results can be attributable to outliers. We appreciate the Reviewer’s recommendation to confirm this.*

Larger Concern 2: The question of causality is somewhat confusing in this paper. Are stressful life events hypothesized to lead to externalizing behavior or the other way around? If stressful life events is the “outcome” and EXT is the “mediator”, you are assuming that one leads to the other. Wouldn’t it be equally plausible that stressful life events could lead to externalizing behavior and acting out? Some sort of theoretical model or textual framing would help here in elucidating the ways that the authors are imaging their conceptual framework. I can see you that

you have a footnote describing this alternative hypothesis, but I wonder if it wouldn't help to discuss more clearly in the intro how you imagine the direction of causality.

RESPONSE: *We may not have been as clear as hoped on the first draft of this manuscript, so allow us to clarify. We do not expect externalizing problems to be a strong mediator of later stressful life events. Rather, we hypothesize that stressful life events predict later externalizing problems more strongly than the reverse. Indeed, this is how we conceptualized the primary model – that is, stressful life events increase vulnerability to psychopathology, and this is especially true in the context of prolonged early adversity. Thus, we completely agree with the Reviewer on this point.*

Having said that, either of these directional effects is plausible: stressful life events could lead to more acting out and dysregulated behavior; or more dysregulated behavior could place individuals at risk for negative life events (perhaps especially for dependent events, over which they have some degree of influence). We elaborate on this point in the Supplementary Materials, where we test a reciprocal cross-lagged model using path analyses. In this analysis, life events and externalizing behavior are tested simultaneously as outcomes and mediators of one another. This analysis, which controls for within-time covariances and cross-time stabilities in life events and externalizing problems, allows us to draw firmer conclusions around the directionality of effects (Adachi & Willoughby, 2015). The results of this analysis are presented in Figure S3 and demonstrate that stressful life events are more strongly predictive of later externalizing problems than the reverse. This analysis also controls for one's history of institutional rearing and gender. As suggested by the Reviewer, our argument indeed assumes a directional relation between life events and psychopathology, and this analysis supports that claim.

We mention this in the Discussion on page 19 when we say: “Fourth, as children were not randomly assigned to high versus low levels of life events, we cannot definitely conclude that stressful life events are causally linked to higher externalizing problems. However, supplementary analysis (Figure S3) showed that more stressful life events at age 12 predicted higher externalizing problems at age 16, while externalizing problems at age 12 only weakly predicted life events at age 16. This effect was independent of children's history of institutionalization. Thus, the relation between life events and problematic behavior appears to be directional in this sample, and cannot simply be attributed to differences in life events across groups” (pg. 19).

Larger Concern 3: Where is the list of other covariates that would have been included? E.g. household wealth, family educational status, size of household, demographic structure of household, type of school child attends, whether parents are married, parental mental health, etc. There are so many variables that would affect both life events and externalizing behavior and it is hard to ignore them here.

RESPONSE: *We appreciate this point and agree that there are multiple potential sources of variance in the current study. Unfortunately, we do not have measurement of family-level socioeconomic factors or family characteristics in the current study. These are all important factors in the wider ecology of children's lives that may be linked to externalizing problems. Our focus in BEIP was comprehensive assessment of developmental outcomes spanning several*

Baseline characteristics of institutionally-reared children

	CAUG (n = 68)	FCG (n = 68)	p-value
Female, No. (%)	35 (51.5)	33 (49.3)	.80
Birth weight (grams), M (SD)	2847 (570.2)	2733 (576.2)	.31
Gestational age (weeks), M (SD)	37.6 (1.5)	37.0 (2.4)	.12
Height for age percentile (cm), M (SD)	26.9 (23.1)	25.7 (22.5)	.78
Weight for age percentile (kg), M (SD)	22.7 (24.6)	18.2 (19.4)	.26
Duration of institutionalization (weeks), M (SD)	87.9 (17.9)	85.2 (23.0)	.47

domains, with consistency in measurement over time. Thus, the depth of measurement is excellent, with some sacrifice on the breadth of measurement across all possible domains of family functioning. While the RCT component takes account of some baseline differences between groups (see Table below, where CAUG and FCG do not differ on any variable), it is still plausible that unaccounted-for differences between the groups explain some of the differences in stress sensitization across the groups. We now mention this as a limitation in the Discussion on page 19.

Other Concerns

Methods: Question about missingness – how are the 134 children included in the study similar/different from the total of 149. The discussion raises this issue as a limitation, but then says it is not an issue because there were still significant findings. The issue, though, is what type of kid you might have lost, and how that would have changed your inferences based on your sample. You could consider IPW (inverse probability weighting) as a sensitivity analysis to make sure that there are no issues with your existing sample being fundamentally different from the 10% who are missing.

RESPONSE: *Thank you for raising this. As you mention, at age 16 there were 149 children who contributed psychopathology data. At age 12 there were 142 who contributed stressful life events data. There were complete data (present on both variables) for 134 children. So there were 15 children (149 - 134) who had psychopathology data at age 16, but no life events data at age 12. The 15 children who did not have life events data did not differ from the 134 with complete data on their level of externalizing problems at age 16 ($t = -.12, p = .91$). Thus, the data are not missing conditional on the primary outcome. Moreover, these groups did not differ on internalizing problems ($t = -.12, p = .90$), birth weight ($t = -.29, p = .78$), or gender or ethnic composition (i.e., proportion Romanian vs. other).*

Finally, if we perform an analysis in which all children are included – including those who are missing life events data – and we use full-information maximum likelihood estimation (FIML) to handle missing data (which easily accommodates the amount of missingness present; Graham, 2009), we continue to observe the same pattern of results: there is an increase in externalizing problems as the number of total life events increases for CAUG ($B = .48, p = .008$) but not FCG ($B = .06, p = .61$). As before, we do not see this same increase for dependent events among the CAUG ($B = .16, p = .47$) or the FCG ($B = .06, p = .65$). In contrast, we do see the increase in

externalizing problems for independent events among the CAUG ($B = .57, p < .001$), but not the FCG ($B = .06, p = .64$). Taken together, we do not believe that the 10% of children missing data on life events who were excluded from the analyses are biasing the present results in a significant way. Again, thank you for raising this issue.

Methods/Results: Stressful Life Events (text & table 1): Were tests of difference among groups examined, and if so what tests were used? The footnote of the table suggested that there were no significant differences. In the text, the authors describe that the items were re-scaled to reduce skew. How were tests of difference conducted with this variable?

RESPONSE: *Yes, differences between groups on each discrete number of life events (i.e., whether they differed on their report of 0 events, 1 event, 2 events, etc.) were assessed from chi-square difference testing using z-tests to compare column proportions (i.e., differences between groups). We have now updated this section of the footnote in Table 1.*

The re-scaling was performed on the count variable (number of life events) for which individuals reported between 0 to 12 life events. As mentioned in the main text, only 8.5% of the children reported between 8 to 12 life events, so we truncated the variable to be 0 to 7+ life events to reduce skew. Group differences on this variable were evaluated using standard ANOVAs with the count variable for total, dependent, and independent events square-root transformed (since they were treated as dependent variables in this analysis). We have updated the caption of Figure S1 to reflect this as well. Thank you for bringing this to our attention.

Methods/Results: There is something a little confusing about Table 1, since it is double counting two sets of children. I wonder if it wouldn't be simpler to present three columns – NIG, CAUG, FCG. It also raises the question of multiple hypothesis testing if you are looking at all of these different comparisons across group. Have you adjusted for multiple hypothesis testing?

RESPONSE: *We understand how this may be confusing. Often, readers are interested in two sets of findings: One is the “institutionalization effect”, which contrasts the ever-institutionalized (EIG) and never-institutionalized (NIG) children; and the second is the “intervention effect”, which contrasts the foster care group (FCG) and care-as-usual group (CAUG). However, we agree with you that, for this Table, the presentation is not as digestible and that the 3-group presentation is more intuitive. We have updated Table 1 to present the results this way.*

Please note that we do not adjust for multiple comparisons in this study, as there are two sets of planned comparisons involving 2 groups each: the EIG vs NIG (institutional rearing effect) and FCG vs CAUG (intervention effect). We now state this clearly on page 10:

“The analyses were conducted in SPSS version 21 and Mplus version 7. Two sets of analyses were carried out for the stress sensitization model. The first analysis ($N = 134$) compared never-institutionalized to ever-institutionalized children, and thus examined differences based on one's history of institutional rearing. The second analysis ($N = 95$) used an intent-to-treat approach to examine whether assignment to foster care was associated with less problematic outcomes compared to care-as-usual (i.e., prolonged institutional care).”

Methods/Results: Externalizing Psychopathology. As presented, this variable sounds like another outcome measure, with the hypothesis that institutionalization would increase externalizing. But the description mentions that “this factor is used as a covariate”.

RESPONSE: *Externalizing problems at age 16 is indeed the outcome variable. The covariate is externalizing problems at age 12 (the prior wave of data collection, concurrent with life events). By controlling for earlier levels of externalizing problems at a time point that is concurrent with life events, the analysis is an auto-regressive model that enhances inferences around the direction of effects (Adachi & Willoughby, 2015). This model is therefore much more conservative than a simple cross-sectional analysis because it controls for past levels of the outcome (externalizing problems) to predict change over time. We have altered the text here to make this more clear (pg. 11 and pg. 12).*

Methods/Results: Dependent and independent life events? What is the difference? This terminology is used in the statistical analysis section but does not appear above in the description of the variable.

RESPONSE: *Consistent with multiple previous studies on stress sensitization, we differentiate between dependent and independent life events conceptually in the last paragraph of the Introduction (pg. 5). Briefly, independent events are those that are not within the child’s control (e.g., a loved one died or had a serious illness), while dependent events are those that the child has at least some degree of control over (e.g., got bad grades in school; got in a fight or broke up with boyfriend/girlfriend). We now mention these two domains in the Methods section (pg. 9), and note that the whole scale (including categorization into dependent and independent events) can be found in the Supplementary Materials.*

Methods/Results: If EXT wasn’t used as an outcome measure, it is unclear how it was used in the model. How does it fit conceptually with the system you are testing?

RESPONSE: *EXT at age 16 is indeed the outcome. This was used as the outcome/dependent variable in all regression analyses, and is the dependent variable plotted in Figure 2. As above, EXT at age 12 was the covariate. We hope the adjustments we have made this clear in the text.*

Results: Not clear what this result means: “this effect was stronger as time spent in institutions increased” Where are these results presented?

RESPONSE: *These results are presented in the Supplementary Materials. Figure S2 shows that the relation between life events and EXT at age 16 is much stronger at higher compared to lower levels of time spent in the institutions. Please let us know if this is still unclear.*

Discussion: Do you have any explanations for why children who went into foster care were receiving “buffering them against stress during periods of significant neural reorganization.” Do you know that this is true from earlier work? Or is this just speculation?

RESPONSE: *Thank you for raising this very important point, as we agree it is worthy of elaboration. Indeed, this was primarily speculation prior to this revision. Both you and Reviewer*

#1 cued us to this issue, and we now offer evidence that we think at least partially explains this result. In short, it is plausible that this “buffering” effect is explained by increased caregiving quality among the foster care versus the care-as-usual children. We copy our response to Reviewer #1 below, which we think addresses this:

“...the difference between the foster care and care-as-usual children may be due to their responsiveness to caregiving experiences during (pre)adolescence, with the foster care children experiencing higher quality care and being buffered against the experience of stressful events to a greater degree than the care-as-usual children. Indeed, we provide new evidence in the Supplementary Materials that children in the foster care group experience higher-quality caregiving at age 12 and 16 years relative to the care-as-usual children ($t = 3.82, p < .001$ at age 12; $t = 3.14, p = .002$ at age 16). Moreover, using Poisson regression we see that, over and above the effects of group (foster care vs. care-as-usual), higher-quality caregiving is associated with lower total (wald $\chi^2 = 15.09, p < .001$), dependent (wald $\chi^2 = 10.03, p = .002$), and independent (wald $\chi^2 = 8.18, p = .004$) life events. So the foster care children experience higher-quality care, on average, compared to the care-as-usual children, and this higher-quality care is related to fewer stressful life events.

Finally, using Poisson regression again, we see that the effect of caregiving quality on stressful life events may be stronger for the foster care versus the care-as-usual children. Specifically, higher quality caregiving is associated with fewer total life events among the care-as-usual children (wald $\chi^2 = 4.39, p = .04$), but this effect is even stronger among the foster care children (wald $\chi^2 = 13.03, p < .001$). We see the same pattern for dependent life events (care-as-usual: wald $\chi^2 = 2.97, p = .085$; foster care: wald $\chi^2 = 8.61, p = .003$) and independent life events (care-as-usual: wald $\chi^2 = 2.99, p = .084$; foster care: wald $\chi^2 = 5.81, p = .016$). From these results, it also appears that the difference between the foster care and care-as-usual children is larger for dependent vs. independent events. Thus, not only do the foster care children experience higher quality care, but higher quality care may exert a stronger protective effect against stressful life events for the foster care children compared to the care-as-usual-children.”

We have now added these results to the Supplementary Materials and provide a brief overview of them in the Discussion (pg. 17-18).”

Thus, we think the buffering effect can be partly attributable to higher quality care that children in the FCG experience, and we proffer this as an area in need of further evaluation using a combination of behavioral and physiological data in future studies (pg. 18).

Minor comments:

Figure 2 could use description of acronyms

RESPONSE: We agree with this and, consistent with the suggestion of Reviewer #1, have removed the majority of study-specific abbreviations for ease of reading.

Might be helpful to have website link for “MacArthur Foundation Research Network for Psychopathology and Development”

RESPONSE: *We have added this website to the Methods section of the paper.*

Once again, thank you to the Editor and Reviewer's for their thoughtful, diligent, and respectful critique of our paper. We believe it is much improved as a result of their feedback and recommendations.

Sincerely,

The authors

Reviewers' Comments:

Reviewer #1:

Remarks to the Author:

I think it is unusual for authors to respond as constructively to comments as have Wade et al. I note in particular 1) the integration of the 'stress-diathesis' implications into the Discussion, 2) the expanded and more thoughtful consideration of the link to HPA function and 3) removing the rather speculative amygdala – PFC text. Likewise the consideration of "dependent" vs "independent" events has added considerably to the paper, as has the resulting additional information. I support acceptance of a very important paper.

Reviewer #2:

Remarks to the Author:

The authors have addressed the concerns that I raised, and I believe this revised version of the manuscript more accurately reflects the study's findings.

Reviewer #3:

Remarks to the Author:

The paper is greatly improved - congratulations! I have no additional comments.

****REVIEWERS' COMMENTS:**

Reviewer #1 (Remarks to the Author):

I think it is unusual for authors to respond as constructively to comments as have Wade et al. I note in particular 1) the integration of the 'stress-diathesis' implications into the Discussion, 2) the expanded and more thoughtful consideration of the link to HPA function and 3) removing the rather speculative amygdala – PFC text. Likewise the consideration of “dependent” vs “independent” events has added considerably to the paper, as has the resulting additional information. I support acceptance of a very important paper.

Reviewer #2 (Remarks to the Author):

The authors have addressed the concerns that I raised, and I believe this revised version of the manuscript more accurately reflects the study's findings.

Reviewer #3 (Remarks to the Author):

The paper is greatly improved - congratulations! I have no additional comments.

RESPONSE: *Thank you again to the Editor and Reviewers for their thoughtful commentary and recommendations for improvement.*